# Profiling Plant circRNAs Provides Insights into the Expression of Plant Genes Involved in Viral Infection

**DOI:** 10.3390/life15071143

**Published:** 2025-07-20

**Authors:** Ghyda Murad Hashim, Travis Haight, Xinyang Chen, Athanasios Zovoilis, Srividhya Venkataraman

**Affiliations:** 1Department of Cell & Systems Biology, University of Toronto, Toronto, ON M5S 3B2, Canada; ghyda.hashim@alumni.utoronto.ca; 2Department of Biochemistry & Medical Genetics, University of Manitoba, 745 Bannatyne Ave, Winnipeg, MB R3E 3N4, Canada; travis.haight@umanitoba.ca (T.H.); athanasios.zovoilis@umanitoba.ca (A.Z.); 3Faculty of Applied Science C Engineering, University of Toronto, 35 St. George Street, Toronto, ON M5S 1A4, Canada; xinyang.chen@mail.utoronto.ca

**Keywords:** circular RNAs (circRNAs), *Arabidopsis thaliana*, Turnip rosette virus (TRoV), Rice yellow mottle virus (RYMV), chloroplast, Gene ontology (GO), photosynthesis, microRNA (miRNA), exogenous circular satellite RNA (scLTSV)

## Abstract

Investigations of endogenous plant circular RNAs (circRNAs) in several plant species have revealed changes in their circular RNA profiles in response to biotic and abiotic stresses. Recently, circRNAs have emerged as critical regulators of gene expression. The destructive impacts on agriculture due to plant viral infections necessitate better discernment of the involvement of plant circRNAs during viral infection. However, few such studies have been conducted hitherto. Sobemoviruses cause great economic impacts on important crops such as rice, turnip, alfalfa, and wheat. Our current study investigates the dynamics of plant circRNA profiles in the host *Arabidopsis thaliana* (*A. thaliana*) during infections with the sobemoviruses Turnip rosette virus (TRoV) and Rice yellow mottle virus (RYMV), as well as the small circular satellite RNA of the Lucerne transient streak virus (scLTSV), focusing on circRNA dysregulation in the host plants and its potential implications in triggering plant cellular defense responses. Towards this, two rounds of deep sequencing were conducted on the RNA samples obtained from infected and uninfected plants followed by the analysis of circular RNA profiles using RNA-seq and extensive bioinformatic analyses. We identified 760 circRNAs, predominantly encoded in exonic regions and enriched in the chloroplast chromosome, suggesting them as key sites for circRNA generation during viral stress. Gene ontology (GO) analysis indicated that these circRNAs are mostly associated with plant development and protein binding, potentially influencing the expression of their host genes. Furthermore, Kyoto Encyclopedia of Genes and Genomes (KEGG) analysis showed photosynthesis as the most affected pathway. Interestingly, the non-coding exogenous scLTSV specifically induced several circRNAs, some of which contain open reading frames (ORFs) capable of encoding proteins. Our biochemical assays demonstrated that transgenic expression of scLTSV in *A. thaliana* enhanced resistance to TRoV, suggesting a novel strategy for improving plant viral resistance. Our results highlight the complexity of circRNA dynamics in plant–virus interactions and offer novel insights into potential circRNA-based strategies for enhancing plant disease resistance by modulating the differential expression of circRNAs.

## 1. Introduction

Circular RNAs (circRNAs) have recently emerged as important regulators of gene expression in various organisms, including plants. circRNAs have been implicated in diverse biological processes, including developmental and stress responses. Previously, they were deemed as an outcome of aberrant splicing with limited functional capability [1]. circRNAs are generated by exon back-splicing from precursor mRNA wherein the upstream 3′ splice site is linked to the downstream 5′ splice site, resulting in an RNA circle produced by a 3′–5′ phosphodiester bond at the linkage site [2,3]. In plants, circRNAs are derived from introns, exons, and intergenic regions of the genomes [4,5]. Their expression depends on the type of tissue, the cell, and their developmental stage. In particular, they are induced in plants under conditions of stress [6,7,8]. In plants, circRNAs function as sponges for miRNAs and are involved in the regulation of precursor RNA alternate splicing [9] and thereby control gene expression. Deep sequencing (RNA-seq) has shown that circRNAs are expressed extensively in plants, including moso bamboo [10], maize [11], tomato [12], cucumber [13], rice [6], and Arabidopsis [8].

circRNAs have also been shown to be involved in plant–virus interactions. Studies by [11] have demonstrated the upregulation of 155 circRNAs and downregulation of 5 circRNAs in maize plants upon infection with maize Iranian mosaic virus, wherein 33 circRNAs were projected to interact with 23 miRNAs. In tomato plants, 83 and 32 circRNAs were expressed, respectively, in plants infected with tomato yellow leaf curl virus (TYLCV) and in the uninfected control plants. When the parent genes generating these circRNAs were silenced, it led to decreased accretion of TYLCV in these plants [14]. Sun et al., 2020 [15] found differentially expressed (DE)-circRNAs and DE-lncRNAs in watermelon plants infected with Cucumber green mottle mosaic virus (CGMMV). It was proposed that competing endogenous RNA (ceRNA) networks were formed in response to stress induced by CGMMV infection. A total of 548 circRNAs were found to be differentially expressed in plants infected with CGMMV. DE-circRNAs were recognized to be putative target miRNA mimics.

Sobemoviruses cause significant infection of economically important plants, resulting in major losses in yields and severely impacting agricultural productivity. Their impacts include decreases in the size and number of seeds, fruits, or other reapable plant parts. Sobemoviruses in general have a relatively limited host range, but some species are capable of infecting plants belonging to multiple families inclusive of monocotyledonous and dicotyledonous species [16] (https://www.dpvweb.net/, accessed on 17 July 2025). Discerning their detrimental impacts and pathways of transmission are critical to develop effective strategies to control them and protect crops from diseases caused by these destructive viruses. The sobemovirus TRoV causes infections in turnip plants leading to necrosis of the veins and petioles in addition to rosetting, leaf twisting, and severe dwarfing [16] (https://www.dpvweb.net/, accessed on 17 July 2025). In swede plants, it causes stunting, resetting, and vein-banding [17]. Known hosts of TRoV are confined to 20 species belonging to Solanaceae, Resedaceae, Compositae, and Cruciferae [16] (https://www.dpvweb.net/, accessed on 17 July 2025). The main natural hosts of TRoV include Brassica rapa and other Brassica species. TRoV is also known to support the replication of the scLTSV in its natural hosts, namely Brassica rapa, Sinapsis arvensis, and Raphanus raphanistrum, but not in Nicotiana bigelovii and Thlaspi arvense, showing that host species significantly control this interaction [18]. Another sobemovirus, RYMV, infects the natural hosts Oryza longistaminata and Oryza sativa (rice) wherein it elicits symptoms such as smaller leaves, reduction in the overall size of the plant, diminished chlorophyll content, disrupted uptake of nutrients, and decreased seed germination [19] (https://www.cabidigitallibrary.org/doi/10.1079/cabicompendium.47658, accessed on 17 July 2025).

Among plants, *A. thaliana* is presently the best-characterized with respect to circRNAs, making this plant an appropriate model system for investigating these RNAs. Moreover, its genome has been well-elucidated and several circRNAs and the genes that encode them are already well-characterized in these plants [20]. Therefore, we chose *A. thaliana* to study the roles of plant circRNAs in the context of viral infection. It was shown by Callaway et al., 2004 that TRoV is capable of infecting *A. thaliana* plants [21]. It has also been demonstrated that *A. thaliana* is a non-host for the RYMV [22].

Thus far, the impacts of the Sobemoviruses TRoV and the RYMV as well as the virusoid scLTSV on plant circRNA profiles have not been explored. In the current study, we investigate the dynamics of circRNA expression in the plant model system *A. thaliana* during viral infections, focusing on TRoV as a host virus, RYMV as a non-host virus, and the scLTSV virusoid. By characterizing changes in circRNA expression profiles and analyzing their potential regulatory functions, we aim to shed light on the intricate interplay between circRNAs and viral pathogens in plants.

We hypothesize that viral RNA influences plant circRNA profiles by potentially modulating regulatory processes, including transcription. We evaluate circRNAs and their predicted impact on the dysregulation of plant cellular genes in response to viral infectious processes. We profile plant circRNAs induced by different plant viruses and circular virusoid RNA, and determine which circRNAs react in response to virus infection, viral components, and circular satellite RNAs. We propose that the induced circRNAs could be potentially involved in viral replication, plant defense mechanisms against viruses, and viral disease symptoms. We catalog other virus/viral component-induced circRNAs involved in regulatory functions such as gene transcription, RNA splicing, etc. We examine the impact of the scLTSV virusoid on the plant endogenous circRNA profile and regulatory processes. We investigate the impact of a non-host virus (RYMV) on the profile and nature of induced plant circRNAs, and finally we determine whether the expression of exogenous circRNAs, such as a virusoid, influences the expression of endogenous circRNAs in the host plant. Our study offers a unique contribution to our understanding of plant circRNAs, particularly regarding their interactions in the context of viral infections.

## 2. Materials and Methods

### 2.1. Plant Propagation, Inoculation with Virus, RNA Extraction, and RT-PCR

*A. thaliana* seeds were pre-germinated on minimal agar plates (0.8%) for 2 days at 4 °C in the dark, followed by growth in a greenhouse under a 16 h light/8 h dark cycle at 23 °C. Seedlings at the 2-leaf stage with root length approximately 1 mm were transferred to soil. Four-week-old *A. thaliana* plants were inoculated with TRoV inoculum, and leaves were harvested at 5, 10, and 15–18 days post-inoculation to track viral systemic movement.

Total RNA was extracted from healthy and infected *A. thaliana* plants under RNase-free reaction conditions as specified in Sambrook and Russell, 2001 [23]. A total of 30 mg of *A. thaliana* leaves was used per RNA sample and each sample was subjected to phenol/chloroform extraction using TES buffer (200 mM Tris pH 7.4 and 25 mM EDTA) followed by ethanol precipitation in the presence of 1/10th volume 3M sodium acetate (pH 5.2) and 2.5 volumes of 95% ethanol (Thermo Fisher Scientific, Waltham, MA, USA). The extracted RNA samples were dissolved in nuclease-free water (Invitrogen, Carlsbad, CA, USA) followed by treatment with DNase for the removal of contaminating genomic DNA as per the following reaction conditions: 15 μL of total RNA, 2.5 μL of 10× reaction buffer, 2 μL of RNase inhibitor (40 units) (Promega, Madison, WI, USA), 1 μL of DNase (2000 units/mL) (New England Biolabs, Ipswich, MA, USA), and nuclease-free water to a final volume of 25 μL. Samples were incubated at 37 °C for 30 min and subsequently the DNase was removed by phenol/chloroform extraction followed by ethanol precipitation of the remaining total RNA. The total RNA was dissolved in nuclease-free water, and the purity of the RNA extracts was estimated by measuring the ratio of A260/280 absorption with a Nanodrop-1000 spectrophotometer (Thermo Fisher Scientific, Waltham, MA, USA). Quality and integrity of the RNA were validated by 1% (*w*/*v*) agarose gel electrophoresis using Tris/borate/EDTA (TBE) buffer. Subsequently, RNA samples were subjected to RT-PCR.

Total RNA samples purified as described above (200 ng) were used as templates for cDNA synthesis. A 10 μM reverse primer specific to the 3′ end of the coat protein ORF of TRoV (Table 1) was used for the reverse transcriptase reaction as follows: samples containing the template and primer mixtures were denatured at 75 °C for 2 min and then quickly chilled on ice for 1–2 min. The following reagents were then added to each sample: 2.5 μL of 10× AMV Reverse Transcriptase (RT) buffer (New England Biolabs, Ipswich, MA, USA), 1 μL of (10 mM) dNTPs (Thermo Fisher Scientific), 1 μL of ribonuclease inhibitor (40 U/μL), and 2.5 μL of AMV RT (200 units) in a 20 μL reaction. The mixture was then incubated for 1 h at 42 °C. Subsequently, 2 μL of the prepared cDNA generated as described above was used as a template for the next PCR step wherein it was subjected to PCR amplification to detect the desired PCR product specific to the virus in question. The PCR reaction contained the following components: 12.5 μL of 2× Taq froggamix (FroggaBio Inc., North York, ON, Canada), 2 picomoles each of reverse and forward primers, 2 μL of cDNA, and nuclease-free water to a final volume of 25 μL. The reaction mix was overlaid with mineral oil and subjected to PCR in a thermal cycler (Applied Biosystems, Thermo Fisher Scientific, Waltham, MA, USA) (Illumina Novaseq ^TM^ 6000) under the following conditions: initial denaturation at 94 °C for 5 min (1 cycle), followed by 35 cycles of denaturation at 94 °C for 45 s, annealing at the specific primer annealing temperature for 45 s, and extension at 72 °C for 1 min, with a final extension at 72 °C for 10 min (1 cycle). After amplification, PCR products were analyzed by gel electrophoresis.

### 2.2. Constructs Prepared in the Current Study

Detailed procedures for construct preparation, including cloning of CP-TRoV, RYMV, and circ-LTSV genes into binary vectors, are provided in the Supplementary Section. Successful cloning for each of the above constructs was confirmed through rigorous validation steps also documented in the Appendix A.

### 2.3. Transformation of Agrobacterium tumefaciens and Generation of Transgenic A. thaliana Plants

The *A. tumefaciens* strain GV3103 was made competent, followed by separate calcium chloride-mediated transformations with the pBI121-CP-TRoV construct, pCambia 1300-RYMV construct, pCambia 1300-scLTSV construct, and empty pBI121 vector control, using kanamycin for the selection of transformants. Specifically, 100 ul of the competent *A. tumefaciens* was incubated with 50 ng of plasmid DNA and the mixture was frozen in liquid nitrogen. All subsequent steps for *Agrobacterium*-mediated transformation were carried out as per Sambrook and Russell, 2001 [23]. Transformed Agrobacterium cells were resuspended in 50 mL solution (5% sucrose, 0.05% Silwet-77 solution). The culture was then sprayed on the young flowers of *A. thaliana*. After four weeks, seeds from these transformed plants were harvested and germinated on agar plates containing Murashige & Skoog (MS) media [24] with Kanamycin for the selection of transgenic plants. The selected transgenic plants were screened for the presence of CP-TRoV or RYMV or scLTSV by total RNA extraction followed by RT-PCR as described above using primers specific to CP-TRoV, RYMV, and scLTSV, respectively (Table 1). Seeds from one line of transgenic *A. thaliana* were selected and screened by germination on 20 μg/mL Hygromycin in 1/2 × MS medium. Germinating plants were considered putative transgenics and were transferred to soil. Resistance to the antibiotic Hygromycin was used as a selection marker to distinguish between transgenic and non-transgenic plants.

### 2.4. Total Plant Protein Extraction, SDS-PolyAcrylamide Gel Electrophoresis (PAGE), and Western Blot Analysis

A total of 0.15 g of *A. thaliana* leaf (fresh) samples were added to autoclaved mortars with liquid nitrogen and ground to a fine powder. The powder was transferred to Eppendorf tubes containing 200 μL protein extraction buffer (50 mM Tris-HCl [pH 7.5], 15 mM MgCl2, 18% glycerol, 0.1 mM PMSF, 7M urea, 2% Triton X-100, and 0.5% NP40). Samples were thoroughly mixed by vortexing and then heated with 5× SDS-PAGE loading dye to 90–100 °C for 7 min. The samples were permitted to equilibrate to room temperature for a few minutes before loading. Estimation of total protein concentrations was performed using a Bradford assay.

SDS-PAGE was conducted using a 4–20% glycine precast gel (Bio-Rad Laboratories, Inc., Hercules, CA, USA), as specified in Sambrook and Russell, 2001 [23]. A volume of 35 μL of each prepared sample, containing between 14.35 µg and 17.15 µg of protein based on the Bradford assay, was loaded into each well. The gel was subsequently subjected to electrophoresis under a constant voltage of 200 V for a duration of 50 min. Subsequent to SDS-PAGE, the protein samples were transferred onto nitrocellulose membrane as per the procedure stated in Sambrook and Russell, 2001 [23]. A 1:1000 dilution of the primary antibody sample raised in rabbits was used, followed by treatment with 1:3000 dilution of the goat anti-rabbit secondary antibody conjugated to alkaline phosphatase. Detection was performed by incubation with alkaline phosphatase substrate solution in the dark for 3–5 min.

### 2.5. Methodology for Observation of Morphological Symptoms of A. thaliana Plants in Response to Virus/Viral Component Infection

Analyses of *A. thaliana* symptoms were conducted to monitor plant growth and discern morphological distinctions among three different groups: healthy, transgenic, and infected plants. All plants were simultaneously grown under the same conditions. First, all seeds were grown on 1/2 × MS medium plates for germination and placed in the growth chamber with 16 h of light and a 22 °C growth chamber program. Once the plants developed roots, they were transferred to soil. Then, the plants were examined under a magnifying stereo microscope to capture images and identify any visible symptoms. Additionally, plant height was measured across six different plants for each sample.

### 2.6. Bioinformatic Analysis

#### 2.6.1. Library Construction and Sequencing by LC Sequencing Company

Total RNA was extracted from *A. thaliana* leaves, and its quantity and purity were assessed using the Bioanalyzer 2100 with an RNA 6000 Nano LabChip Kit (Agilent, Santa Clara, CA, USA). Only samples with an RNA Integrity Number (RIN) greater than 7.0 were used for further processing. Approximately 5 μg of total RNA underwent ribosomal RNA depletion following the Ribo-Zero Plant Removal Kit (Illumina, San Diego, CA, USA) protocol. The remaining RNA was treated with RNase R (Epicentre Inc., Madison, WI, USA) at 37 °C for 30 min to remove linear RNAs, thus enriching for circRNAs (circRNAs). The enriched circRNA fractions were then fragmented using the NEBNext^®^ Magnesium RNA Fragmentation Module (New England Biolabs, Ipswich, MA, USA). The fragmented RNA was reverse-transcribed into cDNA using SuperScript™ II reverse transcriptase (Invitrogen, Waltham, MA, USA). The second strand was synthesized using *E. coli* DNA polymerase I (NEB, USA), RNase H (NEB, USA), and a dUTP solution (Thermo Fisher, Waltham, MA, USA). An A-base was added to the blunt ends to facilitate adapter ligation. Single- or dual-index adapters with T-base overhangs were ligated to the A-tailed DNA fragments, and size selection (300–600 bp) was performed using AMPureXP beads. The ligated products were PCR-amplified under the following conditions: initial denaturation at 95 °C for 3 min, followed by 8 cycles of 98 °C for 15 s, 60 °C for 15 s, and 72 °C for 30 s, with a final extension at 72 °C for 5 min. The average insert size was 300 ± 50 bp. The cDNA library was then sequenced using the Illumina Novaseq™ 6000 platform, generating 2 × 150 bp paired-end reads.

Initial sequencing produced millions of paired-end reads, which were filtered using Cutadapt (version: cutadapt-1.9) to remove adapter sequences, polyA/polyG stretches, reads with more than 5% unknown nucleotides (N), and reads with more than 20% low-quality bases (Q-value ≤ 20). Sequence quality metrics, including Q20, Q30, and GC content, were evaluated using FastQC (version 0.11.9). Filtered reads were mapped to the *A. thaliana* genome using Bowtie2 and Tophat2 (version 2.0.4). Unmapped reads were further analyzed with Tophat-fusion (version 2.1.0) to identify back-splicing junctions indicative of circRNAs. De novo assembly of mapped reads into circRNAs was performed using CIRCexplore2 (version 2.2.6) and CIRI (version 2.0.2). Differential expression analysis was conducted using the R package edgeR. CircRNA expression levels were estimated using the number of back-spliced reads, normalized as spliced reads per billion mappings (SRPBM). Twelve circRNAs were selected for qPCR validation based on their relevance in plant development and virus infection. Each 10 μL qPCR reaction contained 2 μL cDNA (50 ng), 400 nM primer mix, 2 μL nuclease-free water, and 5 μL of 2× Luna Universal qPCR Master Mix (NEB, USA) SYBER green. Real-time RT-PCR was performed to compare expression levels across different plant lines and infection conditions.

#### 2.6.2. Uses of Bioinformatics Software in Identifying circRNAs Relevant to the Current Study

To address the biological questions underlying this study specifically, the identification and functional analysis of circRNAs involved in viral infection and plant defense, a range of bioinformatic tools were employed (Table 2). Quality control was performed using FastQC (version 0.11.9), while Cutadapt (version 1.9) was employed to remove adapter sequences and low-quality reads, thereby ensuring reliable downstream analysis. Sequencing reads were aligned to the *A. thaliana* reference genome using TopHat2 (version 2.0.4) and Bowtie2 (version 2.3.4.1). to support accurate mapping, including the identification of back-splicing junctions, which are crucial for detecting circRNA. Unmapped reads were further analyzed using TopHat-Fusion (version 2.1.0) to enhance the identification of circRNA. CIRCexplorer2 (version 2.2.6) and CIRI (version 2.0.2) were used to annotate circRNAs, helping to uncover circular transcripts that are differentially expressed under viral and transgenic conditions. The edgeR package enabled statistical evaluation of differential circRNA expression across sample groups, including TRoV-infected, satellite-transformed, and control plants.

These tools collectively helped answer key biological questions: Which circRNAs are differentially expressed in response to TRoV infection? Do viral satellite RNAs influence host circRNA expression? Can circRNA profiles distinguish between resistance and susceptibility to infection? By identifying circRNAs specific to certain viral or transgenic treatments, we gained insights into potential regulatory roles of circRNAs in plant immunity. GO enrichment of host genes further clarified the involvement of these circRNAs in pathways related to defense response, stress signaling, and RNA processing. This integrated computational strategy enabled us to link specific changes in circRNA to functional biological processes relevant to plant–virus interactions.

Further, we validated the circular nature of our circRNAs using the divergent primers listed in Table 3.

### 2.7. circRNA Classification Based on Genomic Origins

CIRCexplore and CIRI, renowned circRNA detection tools, were utilized in this study. CIRCexplore employs unmapped reads from Tophat processed through Tophat-fusion (version 2.1) to identify back-splice junction reads, while CIRI2 specializes in datasets containing circRNA species. Both tools were used to detect circRNAs, leveraging their respective strengths. Tophat version 2.0.4 aligned reads to the reference genome (www.plants.ensembl.org, accessed on 17 July 2025), followed by de novo assembly of circRNA reads using CIRCexplore version 2.2.6. Back-splicing junction reads were identified among unmapped reads using TopHat-Fusion and CIRCexplore. Each sample yielded unique circRNAs. Categorization included circRNAs from exonic regions designated as circRNAs, those from intronic regions labeled as ciRNAs, and those from intergenic regions designated as intergenic circRNAs.

## 3. Results Obtained

### 3.1. Symptom Development in A. thaliana Plants Due to Viral Infection

Our investigation began with an assessment of symptom development in *A. thaliana* plants infected with TRoV (host), plants transgenic for the TRoV capsid protein (CP) gene, plants transgenic for scLTSV, and plants transgenic for the RYMV genome (non-host). Observations of plant morphology revealed notable differences between healthy and infected/transgenic plants. CP-TRoV-transgenic plants exhibited reduced height compared to healthy controls (Figure 1), with characteristic black spots on leaf edges. TRoV-infected plants displayed stunted growth and leaf discoloration, progressing from white spots at 10 days post-inoculation (dpi) to purple spots and dry leaves by 15 dpi. Transgenic circ-LTSV and RYMV-infected plants showed no visible symptoms but exhibited variations in plant height compared to controls. Quantitative measurements of plant height for all sample types, including transgenic RYMV, CP-TRoV, circ-LTSV, CP-TRoV-scLTSV, and TRoV-infected plants (10 and 15 dpi), are summarized in Figure 1. Each measurement represents the mean of six biological replicates, and standard deviations were calculated to indicate data variability. Symptom profiles associated with each treatment group highlight the phenotypic effects of complete virus infection versus expression of individual viral genes or virusoids.

#### ANOVA and Tukey HSD Statistical Analyses

We conducted a one-way ANOVA to assess differences in plant height across the seven treatment groups used in our study (RYMV, CP-TRoV, CP-TRoV + scLTSV, Circ-LTSV, Healthy, 10 dpi TRoV, and 15 dpi TRoV). The analysis was based on measurements from six biological replicates per group. The ANOVA results (see the “ANOVA” sheet in the attached Excel file) show a statistically significant difference in mean plant height among the groups (F = 10.02, *p* = 1.9 × 10^−6^), indicating that at least one group differs from the others. We also performed a Tukey HSD post hoc test to determine which specific group comparisons were significant (please find the results in the “Tukey_HSD” sheet in the attached Excel file). For example: CP-TRoV plants were significantly shorter than healthy controls (mean difference = –8.18 mm, *p* = 0.002). 10 dpi TRoV also showed a significant difference from healthy (*p* = 0.014). The raw data used in this analysis are available in the “Raw Data” sheet for transparency. This analysis supports our conclusion that viral treatments, especially CP-TRoV and TRoV infection, have a measurable negative impact on plant growth compared to healthy controls. While the Tukey HSD test compares all possible group pairs, only a few comparisons showed statistically significant differences (marked “True” in the ‘reject’ column). The majority returned “False,” meaning those differences were not significant at *p* < 0.05. These non-significant comparisons are consistent with natural biological variability and group mean overlap. Notably, comparisons like CP-TRoV vs. healthy and 10 dpi TRoV vs. healthy were significant and support our interpretation.

Having observed the above symptoms, we proceeded to investigate the circRNA profiles of these plants to identify their putative molecular interactions in the context of viral infection.

### 3.2. Methodological Confirmations of Clones and Transgenic Plants Used in the Current Study

#### 3.2.1. Detection of CP-TRoV, RYMV, and circ-LTSV Genes in Transgenic *A. thaliana* Plants

Total RNA was extracted from transgenic *A. thaliana* plants expressing CP-TRoV, RYMV, or circ-LTSV genes. RNA integrity was verified, and time-course experiments revealed the early presence of TRoV in the TRoV-infected plants, with RT-PCR confirming viral gene expression.

The following sections describe the separate confirmation of the presence of CP-TRoV, scLTSV, and RYMV transgenes in distinct *A. thaliana*-transgenic plants. This served as the foundation for our further research into the circRNA profiles of these transgenic plants as reported by the current study.

##### Presence of CP-TRoV Gene in Transgenic CP-TRoV *A. thaliana*

To validate the presence of the CP-TRoV transgene in transgenic *A. thaliana* plants, an RT-PCR experiment was conducted (Figure 2). A forward primer (CP-TRoV_F) and reverse primer (CP-TRoV_R) specific to the CP gene were utilized for PCR amplification (Table 1). An 800-base pair (bp) product was detected exclusively in the transgenic plants, indicating the successful integration of the CP-TRoV transgene into the *A. thaliana* genome. In contrast, healthy *A. thaliana* plants (negative control) did not exhibit the presence of the 800 bp band. As a positive control, TRoV-infected *A. thaliana* plants yielded an RT-PCR product at the expected size (800 bp). These results confirm the intact expression of CP mRNA under the control of the 35S CaMV promoter in the pBI121 vector, in which it was cloned. An elongation factor was employed as a loading control to ensure the accuracy of the RT-PCR results.

##### Presence of Genomic RYMV Transgene in RYMV Transgenic *A. thaliana*

An RT-PCR experiment was conducted to verify the presence of the genomic RYMV transgene in transgenic *A. thaliana* plants (Figure 3). A forward primer (CP-RYMV_F) and reverse primer (CP- RYMV_R) specific to the CP-RYMV gene were employed for PCR amplification (Table 1). A 720-base pair (bp) product was detected exclusively in the transgenic plants, indicating the successful integration of the genomic RYMV transgene into the *A. thaliana* genome. In contrast, healthy *A. thaliana* plants (negative control) did not exhibit the presence of the 720 bp band. Similarly, healthy rice plants (negative control) did not demonstrate the presence of the 720 bp band. As a positive control, RYMV-infected rice plants yielded an RT-PCR product at the expected size (720 bp).

##### Confirmation of scLTSV in Transgenic *A. thaliana*

An RT-PCR experiment was conducted to verify the presence of scLTSV in transgenic *A. thaliana* plants (Figure 4). A forward primer (LTSV_sat_F) and reverse primer (LTSV_sat_R) specific to sc- LTSV were utilized for PCR amplification (Table 1). A 322-base pair (bp) product corresponding to scLTSV was detected in transgenic *A. thaliana* plants as well as in positive control samples.

##### Confirmation of Circular scLTSV (circ-LTSV) Replication in Transgenic *A. thaliana* Infected with TRoV

Next, we demonstrated the replication of scLTSV enabled by the helper virus, TRoV. An RT-PCR experiment was conducted to verify the presence of head-to-tail circ-LTSV in transgenic *A. thaliana* plants infected with TRoV (Figure 5). A forward primer (circ-LTSV-F) and reverse primer (circ-LTSV-R) specific to circ-LTSV were utilized for PCR amplification (Table 1). A 322-base pair (bp) product corresponding to circ-LTSV was detected in transgenic *A. thaliana* plants infected with TRoV as well as in positive control samples. Additionally, bands at 522 bp and 844 bp were observed, representing concatamers resulting from the production of circ-LTSV dimer and trimer forms at 322 bp intervals due to the rolling circle replication of scLTSV. The presence of a head-to-tail dimer of sc-LTSV is crucial and a minimum requirement for expressing the circular scLTSV genomic RNA in transgenic plants, facilitated by its self-cleaving ribozyme action through rolling circle replication [25]. Hence, cloning of dimer sc-LTSV was preferred over the monomer form. Elongation factor was used as a loading control to ensure the accuracy of the RT-PCR results.

##### Confirmation of the Absence of CP-TRoV in scLTSV-Transgenic *A. thaliana*

Since the scLTSV-transgenic samples were generated using a construct harboring both CP-TRoV and scLTSV, an RT-PCR experiment was conducted to verify the absence of CP-TRoV in sc-LTSV-transgenic *A. thaliana* plants (Figure 6). A forward primer (CP-TRoV_F) and reverse primer (CP-TRoV_R) specific to CP-TRoV were utilized for PCR amplification (Table 1). No 800-base pair (bp) product corresponding to CP-TRoV was detected from the transgenic plants, indicating the absence of CP- TRoV in the scLTSV-transgenic *A. thaliana*. As expected, positive control samples (TRoV-infected *A. thaliana* plants) showed a band at the expected size (800 bp). Additionally, scLTSV-transgenic *A. thaliana* plants and healthy *A. thaliana* plants (used as negative controls) did not exhibit the presence of the 800 bp CP-TRoV PCR product. Elongation factor was used as a loading control to ensure the accuracy of the RT-PCR results.

#### 3.2.2. Western Blot Analysis

Having confirmed the presence of the CP-TRoV, scLTSV, and RYMV transgenes individually in distinct transgenic *A. thaliana* plants, we proceeded to confirm the detection of CP expression in CP-TRoV-transgenic plants and that of RYMV CP expression in RYMV-transgenic *A. thaliana* plants through Western blot analyses using CP-TRoV antibodies and CP-RYMV antibodies, respectively.

##### Western Blot Analysis for the Detection of CP Expression in Plants Transgenic for CP-TRoV

A Western blot analysis was conducted to evaluate the expression of the CP of TRoV in transgenic *A. thaliana* plants (Figure 7). The results revealed the presence of a distinct 30 kDa CP-TRoV band in the virus-infected positive control sample, demonstrating the specificity and efficacy of the TRoV polyclonal antibody in detecting CP-TRoV expression. The Western blot results indicated that the protein extraction method and Western blotting conditions were optimized for accurate detection. Notably, CP-TRoV-transgenic *A. thaliana* plants also exhibited the CP-specific 30 kDa band, confirming the expression of TRoV CP under the control of the CaMV 35S promoter. In contrast, negative control samples (healthy plants) displayed no equivalent band, further validating the specificity of the CP-TRoV detection.

##### Absence of RYMV Capsid Protein Expression in Transgenic *A. thaliana* as Demonstrated by Western Blot Analysis

A Western blot analysis was conducted to assess the expression of the Rice yellow mottle virus (RYMV) CP in transgenic RYMV *A. thaliana* plants, wherein *A. thaliana* serves as a non-host for RYMV (Figure 8). Protein samples extracted from RYMV-transgenic *A. thaliana* plants and RYMV-infected rice plants (used as a positive control) were subjected to Western blotting. The results revealed a specific CP band at 30 kDa in the protein extract from RYMV-infected rice plants, indicative of CP expression and RYMV genome replication in its natural host. In contrast, the protein extract from RYMV-transgenic *A. thaliana* plants did not exhibit this CP-specific band, suggesting the absence of RYMV CP-protein expression. This observation implies the lack of RYMV replication in transgenic *A. thaliana* plants, consistent with their status as non-hosts for RYMV. Negative control samples, including protein extracts from wild-type *A. thaliana* and healthy rice plants, displayed no detectable CP band, further validating the specificity of the Western blot analysis.

### 3.3. Major Findings from the Current Study

#### 3.3.1. CircRNA Sequencing and Profiling

The advent of high-throughput sequencing (HTS) technology has revolutionized transcriptomic analysis by enabling the detection and characterization of various RNA species, including circRNAs. CircRNAs, being non-polyadenylated RNA molecules, require specialized HTS methodologies for their accurate identification. Notably, the application of linear rRNA depletion strategies, such as RiboZero, has significantly facilitated circRNA sequencing [26]. In this study, we conducted comprehensive bioinformatic analyses to explore the potential functions of circRNAs in *A. thaliana* plants under different conditions. Differential gene expression analysis, functional annotation, and pathway enrichment analysis were performed to elucidate the roles of circRNAs in response to various stimuli. Additionally, we present the results of circRNA sequencing and bioinformatic analyses, followed by detailed sequence mapping.

#### 3.3.2. Results Obtained from RNA-SEQ Bioinformatic Analyses: Identification and Characterization of circRNAs, and Analysis of circRNA Abundance Across Samples and GC Content

Analysis of circRNA sequencing data revealed the generation of more than 40 million valid reads for each sample/run of *A. thaliana* leaves. The GC content was approximately 47%, and the Q30 (quality score) exceeded 92%. Using CIRI2, CIRCexplore, and the R package edgeR (Bioconductor version 3.14 with R version 3.6.3), a total of 760 circRNAs were predicted from the sequencing data. We analyzed the count of circRNAs across all samples, including healthy *A. thaliana*, TRoV-infected plants, and transgenic plants expressing CP-TRoV, scLTSV, and RYMV wherein varying counts were observed.

#### 3.3.3. Genomic Distribution of Identified circRNAs

circRNAs were categorized based on their start and end positions in the genome, distinguishing between exonic, intronic, and intergenic origins. Predominantly, circRNAs originated from exons, with fewer arising from introns, suggesting that mRNA-producing regions are primary sources of circRNA generation. Among the samples, empty vector samples exhibited the highest count of genes generating intronic circRNAs. In contrast, TRoV-infected, circ-LTSV, CP-TRoV, and control samples predominantly produced circRNAs exclusively from exonic or intronic regions. CP-TRoV uniquely produced intergenic circRNAs among the samples.

Figure 9 shows the approximate number of circRNAs originating from the exonic, intronic, and intergenic regions of the plant genome. For instance, RYMV exhibits approximately 16 SRPBM circRNAs from exonic regions and around 3 circRNAs from intronic regions.

#### 3.3.4. Principal Component Analysis

All samples underwent circRNA sequencing with two replicates, and comparative analysis revealed no significant differences between the two sequencing runs for all nine samples. Principal Component Analysis (PCA) was conducted on the normalized RNA-seq data encompassing 760 differentially expressed circRNAs across all nine samples. The resulting PCA plot provides a comprehensive representation of the conditions for all samples, demonstrating consistent representation across the two sequencing runs (Figure 10) and is instrumental in distinguishing negative controls (healthy, EV, and mock-inoculated samples) from the remaining samples within both batches. Overall, the plot provides a comprehensive representation of the conditions for all samples, demonstrating their consistent representation across the two sequencing runs.

#### 3.3.5. Chromosomal Distribution of circRNAs

Analysis revealed circRNA distribution originating from across chromosomes 1 to 5, with a limited presence in mitochondria (Figure 11). Notably, the chloroplast chromosome expressed the highest number of circRNAs.

#### 3.3.6. Diversity in circRNA Splicing Signals

The splicing signals at circRNA splice sites in our study exhibited diversity. Unlike the typical GT- AG splicing signal observed in animals, our sequencing results in *A. thaliana* presented a variety of splicing signals, such as TA/GT and TT/GT, expressed in terms of AGCT due to the DNA sequencing method used (Figure 12).

#### 3.3.7. Patterns of circRNA Counts

In this study, circRNA counts in each sample were analyzed using the R program version 3.6.3 package ggplot2. Our data revealed diverse circRNA counts across different samples. For instance, ciRNA9 exhibited 30 counts at 15 dpi, decreasing to 10 counts at 10 dpi, while the mock-inoculated sample showed 60 counts. Another circRNA, circRNA2, had 23 counts in the non-host sample (RYMV transgenic *A. thaliana*) and 100 counts in the host sample (CP-TRoV transgenic *A. thaliana*). Similarly, ciRNA11 showed varied counts across different samples, with the highest count in the CP-TRoV-scLTSV double-transgenic *A. thaliana*. All samples exhibited zero counts of ciRNA17, while healthy controls showed a high count for this circRNA.

#### 3.3.8. Gene Ontology (GO) Functional Analysis

GO enrichment analysis was conducted for host genes encoding dysregulated circRNAs. Most host genes were associated with plant development and protein binding activity, suggesting a regulatory role of circRNAs in host gene regulation.

#### 3.3.9. Characteristics of Chloroplast-Derived circRNAs

In our study, the chloroplast chromosome exhibited the highest number of identified circRNAs, with a total of 288 circRNAs identified. To explore the diversity of these circRNAs, a subset was selected for further analysis. Notably, six circRNAs from this subset shared the same end position but differed in their start positions on the chloroplast chromosome. For example, ciRNA101 showed five counts in the circ-LTSV transgenic sample, while all other samples, including the control, showed zero counts. Similarly, ciRNA103 had three circRNA counts compared to the control’s circRNA counts. Despite these variations, all circRNAs shared the same ending position and differed by a few nucleotides in the start position (Figure 13). Furthermore, selected circRNAs derived from the chloroplast were chosen for qPCR validation (Figure 14). All selected circRNAs from the chloroplast for qPCR validation exhibited distinct start and end positions without any overlap.

#### 3.3.10. Internal ORF Identification in circRNAs

Variations in circRNA2 counts were observed among transgenic and virus-infected groups. This circRNA exhibited elevated counts in all transgenic samples and reduced counts in TRoV-infected samples compared to the control group. Further analysis revealed that circRNA2 contains an open reading frame (ORF) with start and stop codons, suggesting its potential translatability to a protein. Similarly, circRNA205 (ATCG00020) showed count differences between non-host (RYMV) and host virus samples. Notably, it exhibited 20 counts in the RYMV sample, while zero counts were recorded in the control and CP-TRoV/scLTSV transgenic samples. Furthermore, circRNA205 contained an ORF region in its sequence, indicating potential translatability to a protein with similarities to a wheat protein known for monooxygenase activity [27].

#### 3.3.11. Functional Insights of Identified circRNAs

This study profiled circRNAs from *A. thaliana* in the presence of both host and non-host viruses for the first time. Several key findings include the following:

AT3G60240 (eIF4G): This gene generated circRNA expressed exclusively in the RYMV-transgenic sample. The eIF4G protein, essential for translation initiation, reportedly also influences TCV multiplication [28].

AT4G21790 (TOM1): This gene generated circRNA expressed in the RYMV transgenics. TOM1 codes for a host factor necessary for TMV virus replication [29].

AT3G05500 (SRP3): This gene showed low-level expression in the CP-TRoV-transgenic sample compared to the negative control. Reportedly, overexpression of SRP3 results in improved drought stress tolerance and increased reproductive growth [30].

AT5G37475: This gene was expressed at low levels in the RYMV-transgenic sample. It is known to be involved in the formation of the eIF4G complex, which is crucial for initiating a subset of mRNAs involved in cell proliferation [31]

#### 3.3.12. CircRNA–miRNA Interactions

In this study, we investigated whether identified circRNAs could act as miRNA sponges. Dysregulated circRNAs were predicted as miRNA target mimics using Targetfinder and Miranda. CircRNAs were found to have one or multiple decoy sites for miRNAs (Figure 15). For example, circRNA291 (AT1G01520) might act as a decoy for miRNA ath-miR396b-5p. Similarly, circRNA236 (AT5G10550) might act as a decoy for two different miRNAs (ath-miR167a-5p and ath-miR167b). Multiple circRNAs from the same parental gene (AT1G01520) might act as decoys for one miRNA, such as circRNA ATH_circ04122, ATH_circ04121, and ATH_04124, all of which had miRNA ath-miR396b-5p binding sites.

Further examination revealed that mRNAs in the regulatory network of circRNA-miRNA-mRNA were mainly involved in plant growth and development. This suggests that circRNAs in plants respond to various stress conditions, including biotic and abiotic stresses, affecting miRNA expression profiles. While the circRNA-miRNA-mRNA network has been identified in plants, biochemical analyses are needed for further elucidation. Only one study reported the role of circRNA in plants, where exonic circRNA in *A. thaliana* was shown to regulate the splicing of its parental mRNA [9].

#### 3.3.13. Enrichment Analysis

Enrichment analysis for all dysregulated genes was conducted using ShinyGO to extract essential biological processes, molecular functions, and KEGG pathways. Gene ontology and domain enrichment analyses were performed to elucidate the main biological functions of the identified circRNAs in our dataset. The analysis revealed expected biological functions, including plant development, RNA processing, and protein binding, with mRNA and RNA binding being the most significantly enriched functions (Figure 16 and Figure 17).

#### 3.3.14. Validation of Identified circRNAs by RT-qPCR

To confirm the origin of selected circRNAs, RT-PCR followed by real-time quantitative qPCR was conducted. Primers were designed using Primer 3 software, ensuring the detection of authentic circRNAs with divergent primers and linear RNAs with convergent primers. Divergent primers were specifically designed and used only for a selected subset of circRNAs to validate their circular nature [Table 3].

RNase R treatment was applied to remove linear RNAs, and primer efficiency was validated before qPCR. Efficient primers (90–110% efficiency with R2 > 0.98) were selected for qPCR validation. Gene expression changes observed in qPCR aligned with RNA-seq analysis, confirming the dysregulation of selected circRNAs (Figure 18). Strong correlation between qPCR and RNA-seq data validated the consistency of gene regulation patterns identified across both techniques.

## 4. Discussion

CircRNAs, emerging as vital gene regulators, encompass exogenous and endogenous types, yet their functions remain incompletely understood [32]. While some roles, like miRNA “sponging,” are recognized, much remains unclear, making differential expression analysis valuable [33]. Our study sheds light on circRNA expression during virus infection and plant defense mechanisms. The escalating agricultural impact of plant viruses underscores the significance of understanding circRNA involvement. We explored circRNA profiles in virus-infected, uninfected, and transgenic plants expressing viral genes, revealing differential expression patterns across groups.

To elucidate *A. thaliana’s* response to specific viral genes and non-host viruses, we employed transgenic plants. This novel approach unveils endogenous circRNA landscape alterations, providing insights into plant–virus interactions previously unexplored. Predictive analysis identified abundant and low-abundance circRNAs across experimental groups, revealing diverse origins in the *A. thaliana* genome. Dysregulated circRNAs, mainly from coding exons, underscored their potential regulatory roles in plant cells. GO enrichment analysis highlighted circRNA-associated host genes’ involvement in diverse cellular processes, including stress response and mRNA binding. These findings suggest circRNAs’ regulatory roles in plant defense mechanisms. Distinct circRNA expression patterns in virus-infected and transgenic plants point to potential plant genes implicated in infection processes. Notably, certain genes showed differential expression only in non-host plant samples, suggesting unique plant–virus interactions. PCA revealed distinct circRNA expression patterns among experimental groups, validating data reliability and highlighting circRNA alterations in response to viral infections. Dysregulated circRNAs identified in transgenic plants suggest exogenous circRNA (scLTSV) effects on endogenous circRNA expression, warranting further investigation into underlying mechanisms. Our data showed TA/GT and TT/GT as the highest splicing signals for the circRNAs, while Zhang et al. 2020 [34] reported diverse splicing signals, with CT/AC and GT/AT being prominent. Another study on plant circRNAs, focusing on tomato plants under temperature stress [35], found GU/AG as the predominant splice signal. Identification of dysregulated circRNAs with miRNA binding sites suggests potential miRNA sponging activity, implicating circRNAs in post-transcriptional gene regulation.

In our study, we found that circRNA2, involved in oxidoreductase and cytochrome complex assembly, showed varied expression patterns across different experimental conditions. Interestingly, it was upregulated in samples from RYMV-, circ-LTSV-, and CP-TRoV-transgenic plants but downregulated in TRoV-infected samples compared to controls. The induction of circRNA2 seems to be triggered by various stressors on the plant, including mechanical stress induced by water inoculation and expression of scLTSV through the 35S promoter. This induction mechanism may involve RNA-RNA or RNA-DNA interactions, as scLTSV lacks coding capacity but can upregulate circRNA2, warranting experimental validation. Our findings suggest that transgenic circ-LTSV *A. thaliana* may induce resistance to helper virus TRoV infection, aligning with previous studies on viral satellite RNAs eliciting resistance to viral infections [36,37]. Additionally, several cellular circRNAs, such as ATCG00030 (ciRNA106) and ATCG00130 (ciRNA11), are induced solely by the non-coding exogenous circLTSV. This is a significant result and implicates the potential involvement of uniquely expressed circRNAs in conferring the resistance to TRoV infection.

Enriched pathways associated with photosynthesis and chloroplast-related processes in dysregulated circRNAs indicate intricate virus–chloroplast interactions influencing plant–virus dynamics. The exploration of chloroplast-derived circRNAs in our study revealed intriguing characteristics, particularly regarding their differential regulation in circ-LTSV-transgenic plants. This selective disruption suggests a potential regulatory role for circ-LTSV in chloroplast-derived circRNAs. Additionally, the targeted approach for qPCR validation of circRNAs provided valuable insights into their structural diversity and potential functional roles, indicating nuanced regulation possibly associated with chloroplast-related processes or responses. The dysregulation of circRNAs across various samples, including TRoV-infected plants, RYMV (non-host) plants, and plants expressing viral genes, suggests complex interactions between viral components and plant circRNA pathways. While the exact mechanisms remain to be elucidated, hypotheses such as RNA-RNA interactions, miRNA sponging activity, or sequence similarities with plant miRNA or circRNA may contribute to circRNA dysregulation.

Our study’s findings on circRNA expression patterns, especially in the context of viral infections and transgenic influences, shed light on the intricate regulatory landscape of circRNAs in plants. The genomic distribution of circRNAs and their differential expression patterns across samples provide valuable insights into the potential roles of circRNAs in plant biology and response to viral infections. The observations on plant morphology further underscore the impact of different viral infections on *A. thaliana* plants, with notable differences in growth and symptom progression among TRoV-infected, RYMV-transgenic, and CP-TRoV-infected plants. These findings contribute to our understanding of the interactions between viruses and their plant hosts, potentially informing strategies for managing viral diseases in crops. Unlike other studies that analyze mRNA to detect changes in circRNAs [11,15], our research focused exclusively on circRNAs. This approach allowed us to explore specific circRNAs that lack complementary sequences to miRNA and mRNA, suggesting a novel regulatory mechanism potentially involving RNA-RNA or RNA-DNA interactions, especially in circRNAs dysregulated in transgenic plants expressing the non-coding virusoid circ-LTSV. Remarkably, our findings reveal that the majority of these circRNAs are localized to the chloroplast chromosome and intricately linked to the photosynthesis pathway, as evidenced by KEGG analysis. This association suggests a significant role for circRNAs in regulating critical photosynthetic processes, including light capture and energy conversion. This research not only deepens our comprehension of the functional dynamics of circRNAs within plant virology but also lays the groundwork for novel agricultural biotechnological applications aimed at enhancing crop resilience to viral threats, thus supporting food security.

Beyond viral infections, circRNAs respond to abiotic stresses, such as heat and dehydration [35,38], and other biotic stresses, including infection by *Pseudomonas syringae* and *Botrytis cinerea* in *A. thaliana* [39]. These findings highlight circRNAs as potential regulators of plant immunity. Our study builds on this foundation, using circRNA profiling to identify regulatory networks associated with virus susceptibility and resistance in *A. thaliana*. The majority of circRNAs identified in our dataset originate from coding exons, consistent with other plant studies [34,39].

While previous studies have identified differential expression of circRNAs in plants under viral [11,15], abiotic [35], and other biotic stresses [39], these were often limited to identifying expression changes or predicting miRNA interactions. Most of these studies focused on crops like maize, tomato, and rice, and lacked mechanistic insight into how circRNAs influence plant defense. In contrast, our study utilizes *A. thaliana* as a well-characterized model to explore not only the changes in circRNA expression upon TRoV infection, but also how these circRNAs may regulate gene networks associated with resistance or susceptibility. Furthermore, we investigate the potential role of viral satellite circRNAs, an unexplored area, primarily to understand their impact on host defense mechanisms. This approach provides a more comprehensive understanding of circRNA function in the context of plant–virus interactions.

This study identified distinct circRNA profiles in *A. thaliana* under host (TRoV) and non-host (RYMV) viral conditions. Notably, circRNA296 from AT3G60240 (eIF4G) was expressed only in RYMV-transgenic plants, suggesting that viral RNA alone, without viral protein expression, can induce specific host translation factors. This contrasts with findings in rice, where RYMV infectivity requires VPg eIF(iso)4G1 interaction [40]. Since VPg is translated from subgenomic RNA, its absence in *A. thaliana* may explain the non-host incompatibility. Another translation-related gene, AT5G37475, was also more abundant in RYMV-transgenic plants, reinforcing the potential role of eIF4G complexes in non-host resistance. Prior studies have shown that VPg eIF4E/iso4G interactions are essential for infectivity in several plant viruses [41,42].

While many studies have reported microRNA (miRNA) regulation in response to viral infection, few have explored circRNA dynamics [11,15]. Also, no systematic analysis has addressed how viral satellite circRNAs affect plant defense mechanisms. Our findings regarding circ-scLTSV transgenic resistance to TRoV may involve the regulation of translation factors by this circRNA virusoid. Importantly, the observed changes were not linked to general stress responses, indicating virus-specific regulation.

Overall, this study provides extensive insights into the role of circRNAs in plant–virus interactions, highlighting their potential as targets for further research in plant biology, virology, and agricultural biotechnology.

## 5. Conclusions

In conclusion, this current study provides novel insights into the regulatory role of circRNAs in plant–virus interactions, particularly in *A. thaliana*. Our findings highlight the diverse functions of circRNAs in modulating host gene expression and cellular responses to viral infections. We propose that further investigation into circRNA-mediated mechanisms could lead to the development of innovative strategies for enhancing plant resistance to viral diseases, thereby contributing to sustainable agriculture and food security.

## Figures and Tables

**Figure 1 life-15-01143-f001:**
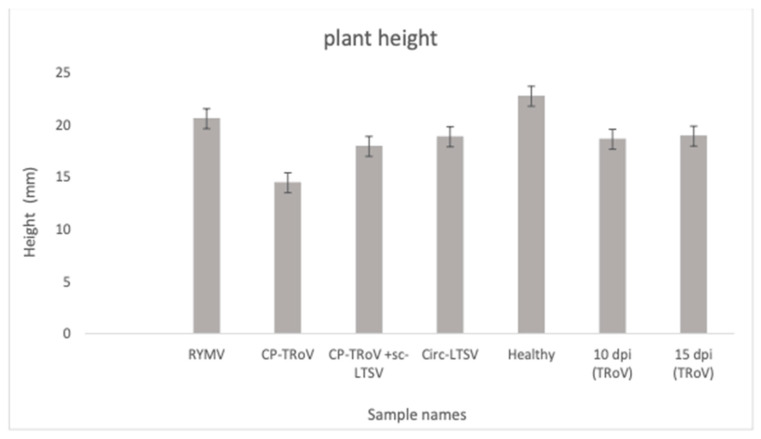
Plant height in all *A. thaliana* samples utilized in this study. The error bars reflect the standard deviation (*n* = 6). mm stands for plant height. Groups include healthy and virus-infected or transgenic A. thaliana. Statistical analysis was performed using one-way ANOVA and Tukey HSD.

**Figure 2 life-15-01143-f002:**
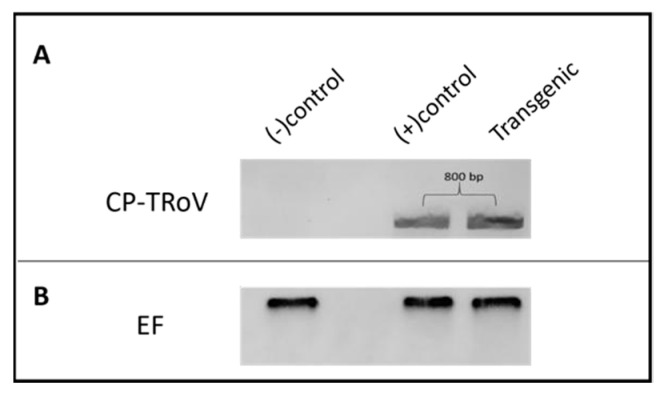
RT-PCR results of genome (virus-infected plants) and transcripts of TRoV CP gene in transgenic *A. thaliana* plants. RNA samples were used after DNase treatment for the cDNA synthesis, followed by PCR. (**A**). Lane 1: RT-PCR product of healthy *A. thaliana* used as the negative control. Lane 2: empty lane. Lane 3: 800 bp RT-PCR product from plants infected with TRoV (positive control). Lane 4: 800 bp RT-PCR product from CP-TRoV-transgenic plants. (**B**). Elongation factor gene was included for internal control experiments. In total, 35 cycles for each analysis were applied.

**Figure 3 life-15-01143-f003:**
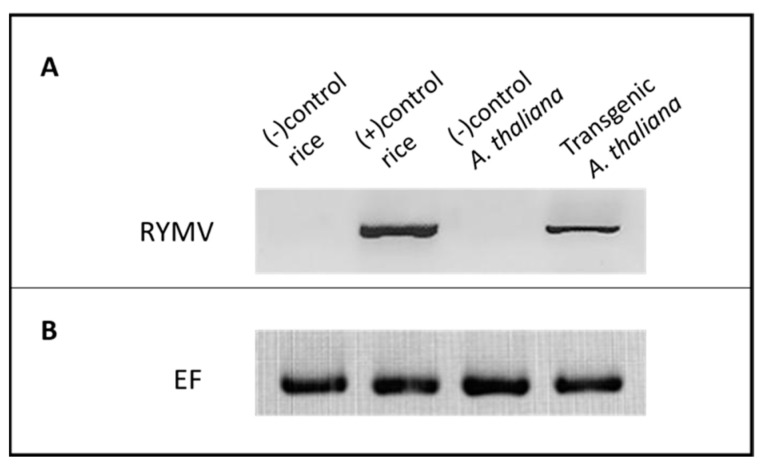
RT-PCR results of genome (virus-infected plants) and transcripts of genomic RYMV in transgenic *A. thaliana* plants. RNA samples were used after DNase treatment for the cDNA synthesis, followed by PCR. (**A**). Lane 1: RT-PCR product of healthy rice used as the negative control. Lane 2: 720 bp RT-PCR product from rice plants infected with RYMV (positive control). Lane 3: RT-PCR product of healthy *A. thaliana* used as the negative control. Lane 4: 720 bp RT-PCR product from genomic RYMV expressed in transgenic *A. thaliana* plants. (**B**). Elongation factor gene was included for internal control experiments. In total, 35 cycles for each analysis were applied.

**Figure 4 life-15-01143-f004:**
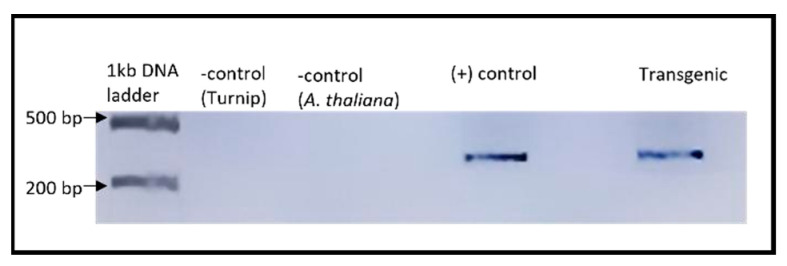
RT-PCR results of sc-LTSV in transgenic *A. thaliana* plants. RNA samples were used after DNase treatment for the cDNA synthesis, followed by PCR. Lanes 1 and 2: No bands were visible in the negative controls. Lane 4: 322 bp product was obtained from turnip infected with LTSV used as a positive control. Lane 6: 322 bp product was obtained in the transgenic *A. thaliana*.

**Figure 5 life-15-01143-f005:**
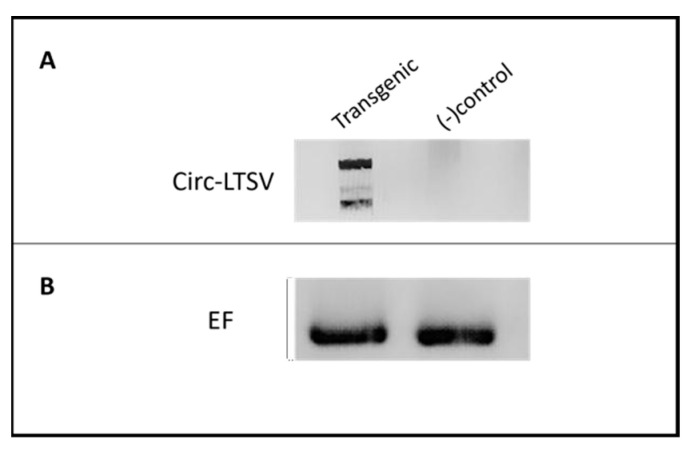
RT-PCR results of sc-LTSV replication in transgenic *A. thaliana* plants infected with TRoV. RNA samples were used after DNase treatment for the cDNA synthesis, followed by PCR. (**A**). A 322 bp product was obtained in the transgenic *A. thaliana* infected with TRoV in addition to other bands that were observed at 522 bp and 844 bp due to the production of concatemers in the rolling circle replication process in lane 1. Lane 2: No bands were visible in the negative control. (**B**). Elongation factor gene was included for internal control experiments. In total, 35 cycles for each analysis were applied.

**Figure 6 life-15-01143-f006:**
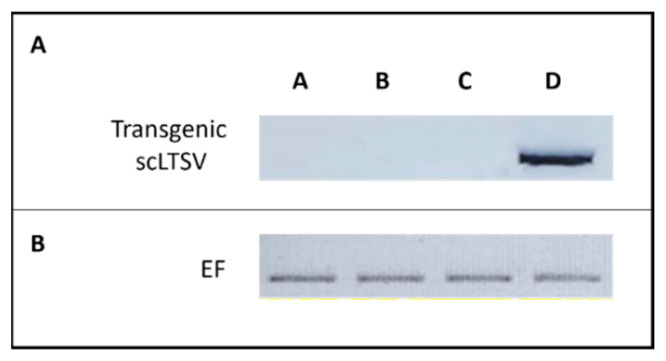
RT-PCR results of transgenic scLTSV *A. thaliana* using CP-TRoV primers. RNA samples were used after DNase treatment for the cDNA synthesis, followed by PCR. (**A**). Lane A: RT-PCR product from the transgenic scLTSV *A. thaliana* used as the negative control. Lane B: RT-PCR product of healthy *A. thaliana* used as the negative control. Lane C: transgenic scLTSV *A. thaliana* infected with TRoV showing the resistance of the scLTSV-transgenic plants to TRoV. Lane D: 800 bp RT-PCR product from *A. thaliana* plants infected with TRoV alone (positive control). (**B**). Elongation factor gene was included for internal control experiments. In total, 35 cycles for each analysis were applied.

**Figure 7 life-15-01143-f007:**
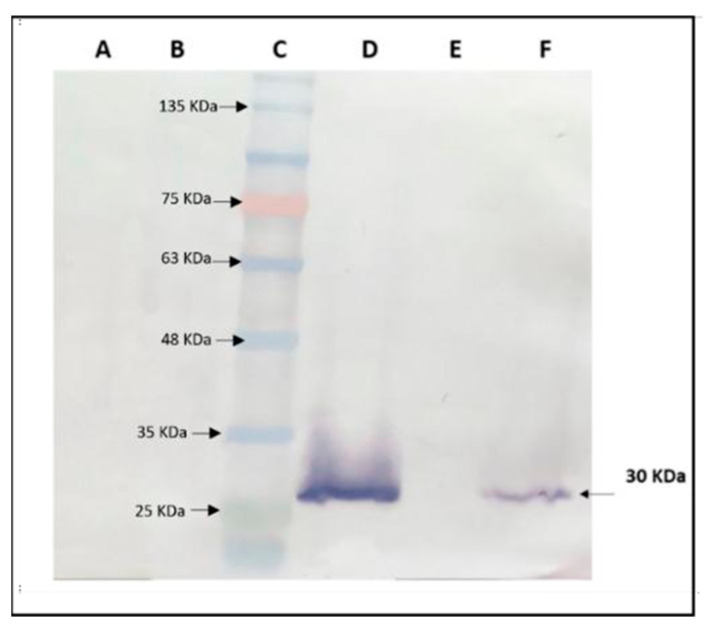
Western blot analysis for detecting CP-TRoV expression in transgenic *A. thaliana*. Lane A: Negative control uninfected, healthy *A. thaliana*. Lane B: Negative control *A. thaliana* harboring the empty binary vector (pBI121), showing no detectable bands. Lane C: Protein molecular weight ladder. Lane D: *A. thaliana* infected with TRoV, demonstrating the abundance of the 30 kDa CP expression. Lane E: Empty lane. Lane F: Transgenic *A. thaliana*, showing TRoV CP as a faint band at 30 kDa. Equal amounts of proteins (~14–17 ug) were added in all lanes following protein estimation using the Bradford assay (see Appendix A).

**Figure 8 life-15-01143-f008:**
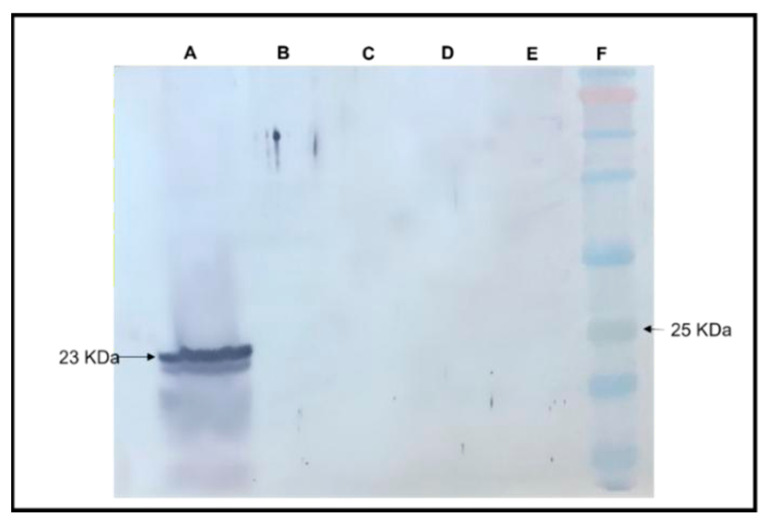
Western blot analysis for detecting RYMV expression in transgenic *A. thaliana*. Lane A: Rice plant infected with RYMV used as a positive control (inoculum) showing the abundance of the 23 kDa CP-RYMV expression. Lane B: Transgenic *A. thaliana* showing the absence of the CP-RYMV at 23 KDa. Lane C: Empty lane. Lane D: Negative control *A. thaliana* showing no detectable bands. Lane E: Uninfected, healthy rice as a negative control. Lane F: Protein molecular weight ladder. Equal amounts of proteins (~14–17 ug) were added in all lanes following protein estimation using the Bradford assay (see Appendix A).

**Figure 9 life-15-01143-f009:**
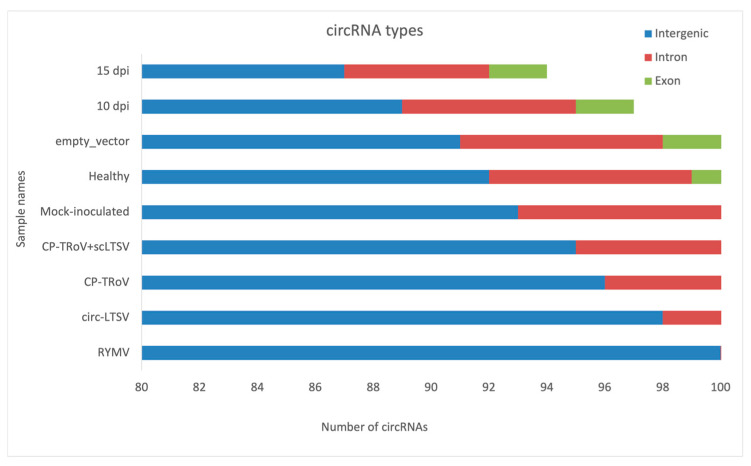
Numerical distribution of circRNAs across coding and non-coding regions in the nine sample groups analyzed in this study. These groups include (1) healthy *A. thaliana* (negative control), (2) mock-inoculated *A. thaliana*, (3) empty vector-transformed *A. thaliana* (pBI121), (4) TRoV-infected *A. thaliana*, (5) transgenic *A. thaliana* expressing CP-TRoV, (6) transgenic *A. thaliana* expressing RYMV, (7) transgenic *A. thaliana* expressing circ-LTSV, (8) transgenic *A. thaliana* co-expressing CP-TRoV and scLTSV, and (9) healthy rice as an external control. Notably, only the CP-TRoV sample showed expression of circRNAs localized in the intergenic region.

**Figure 10 life-15-01143-f010:**
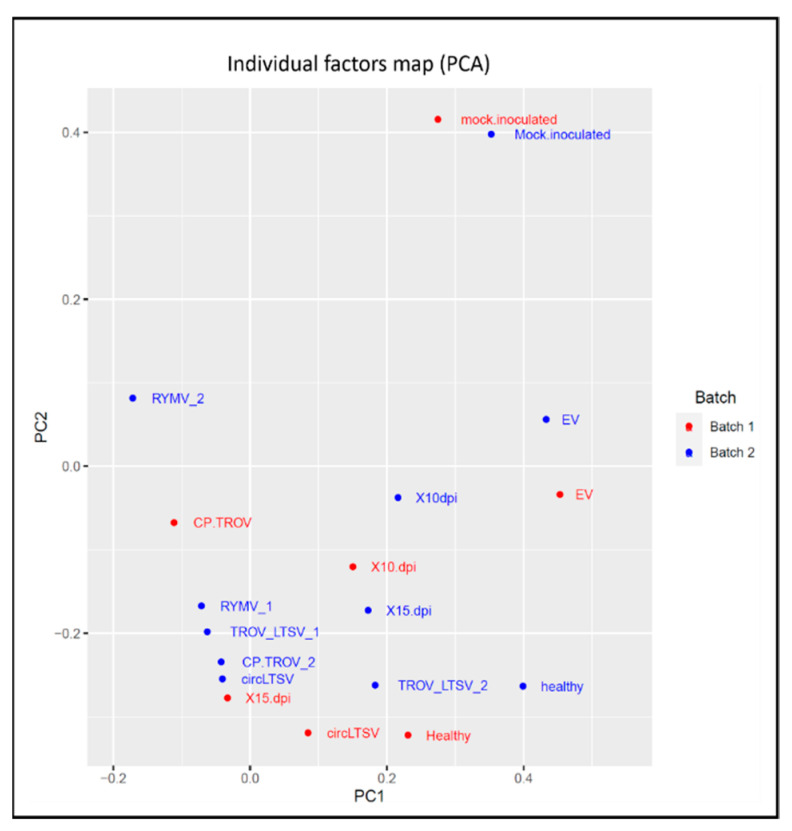
PCA plot of RNA-seq data showing the characteristics of all the samples according to gene expression. The PCA effectively distinguishes between the negative controls and the remaining samples. PC1 refers to the 1st run and PC2 refers to the 2nd run.

**Figure 11 life-15-01143-f011:**
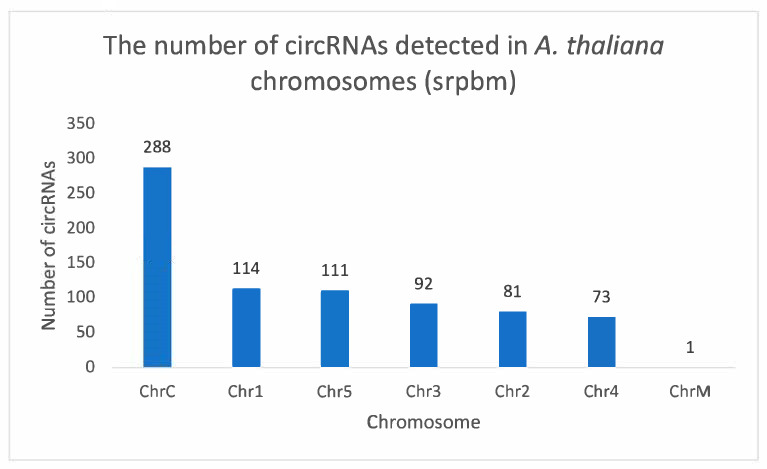
Distribution of circRNAs. Histogram depicting the numerical distribution of circRNAs detected in *A. thaliana* chromosomes (srpbm: spliced reads per billion mappings). “ChrC” refers to the chloroplast chromosome and “ChrM” refers to the mitochondrial chromosome.

**Figure 12 life-15-01143-f012:**
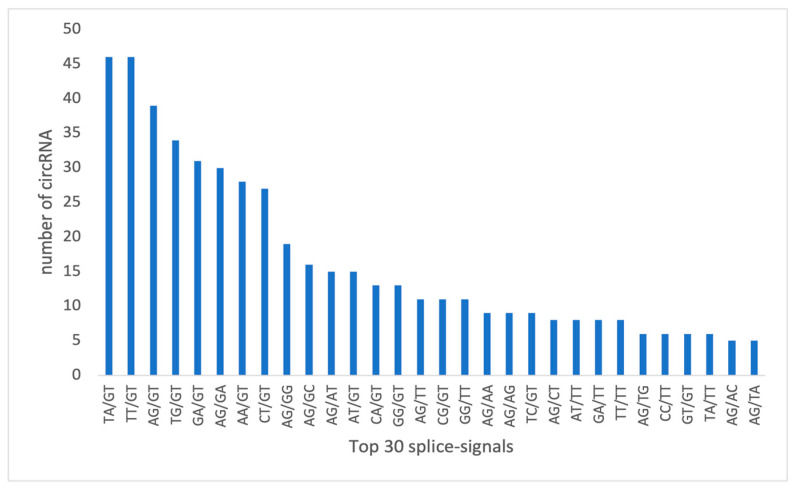
The top 30 splicing signals of circRNAs identified in our circRNA-seq datasets. The histogram displays the most frequent splicing signals observed. Splice site sequences are shown in AGCT format, corresponding to the DNA sequences obtained during library preparation and sequencing.

**Figure 13 life-15-01143-f013:**
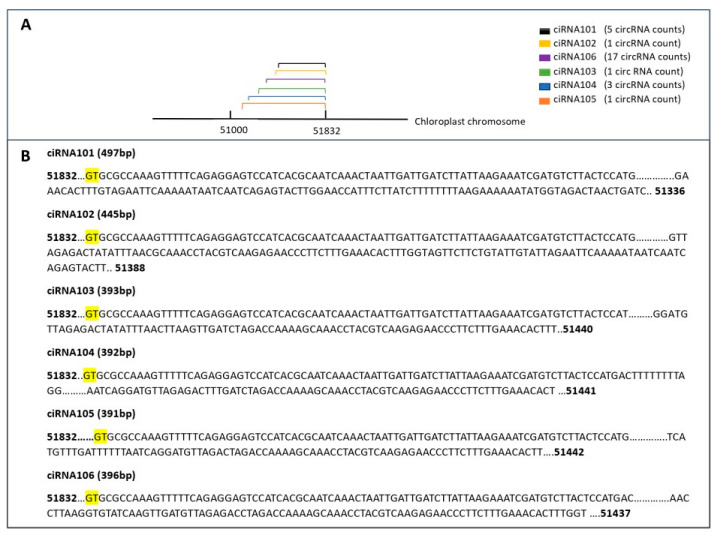
The variation in a selected group of circRNA positions on the chloroplast chromosome. (**A**) features the chromosome line and circRNA lines, providing a clear representation of circRNA positions. (**B**) illustrates circRNAs on the chloroplast sequence, emphasizing each circRNA’s start position while maintaining a consistent end position. The junction (splice site) is highlighted in yellow.

**Figure 14 life-15-01143-f014:**
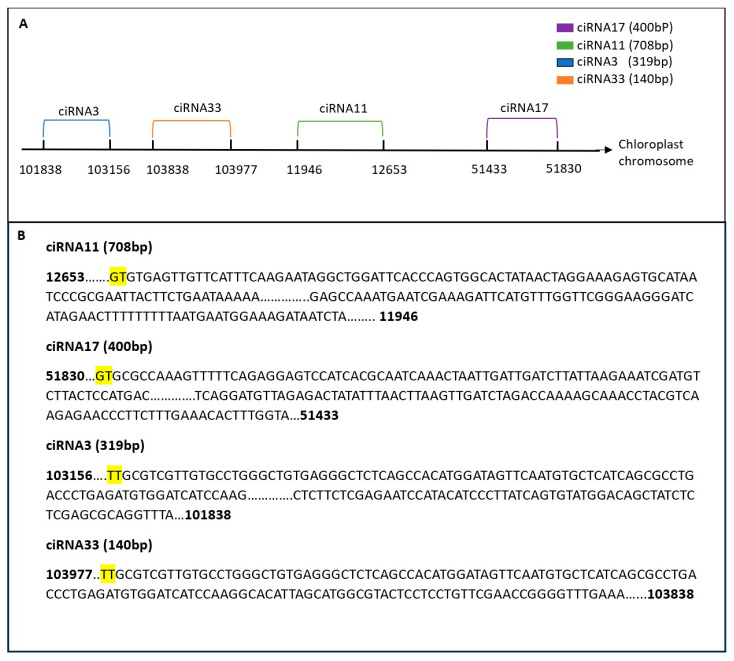
CircRNA selection on the chloroplast chromosome for qPCR validation. In (**A**), selected circRNAs on the chloroplast chromosome are represented by lines indicating their start and end positions, without any overlap. (**B**) displays the sequence of each circRNA, emphasizing the splicing signals. The junction (splice site) is highlighted in yellow.

**Figure 15 life-15-01143-f015:**
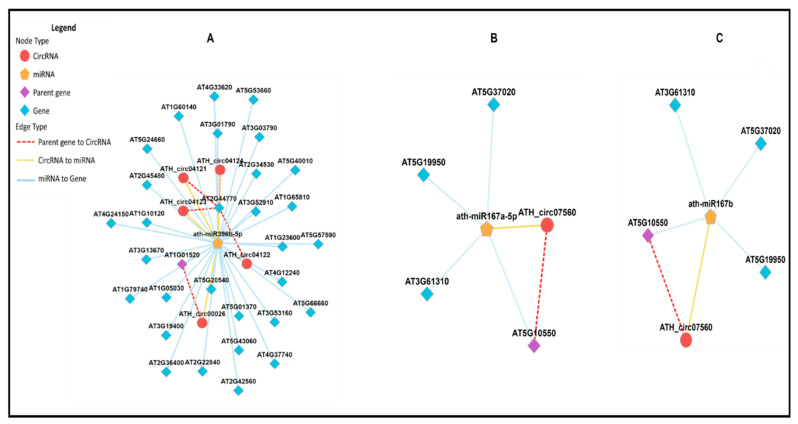
The circRNA-miRNA-mRNA interaction network. Yellow nodes: miRNAs. Red nodes: circRNAs that may be miRNA decoys. Blue nodes: mRNAs (genes) that may be miRNA targets. Purple nodes: parental genes. (**A**) shows the networking connection for multiple circRNAs from the same parent gene. (**B**,**C**) represent the same circRNA targeting two miRNAs. The public plantCircNet database was used to construct the circRNA-miRNA-mRNA network.

**Figure 16 life-15-01143-f016:**
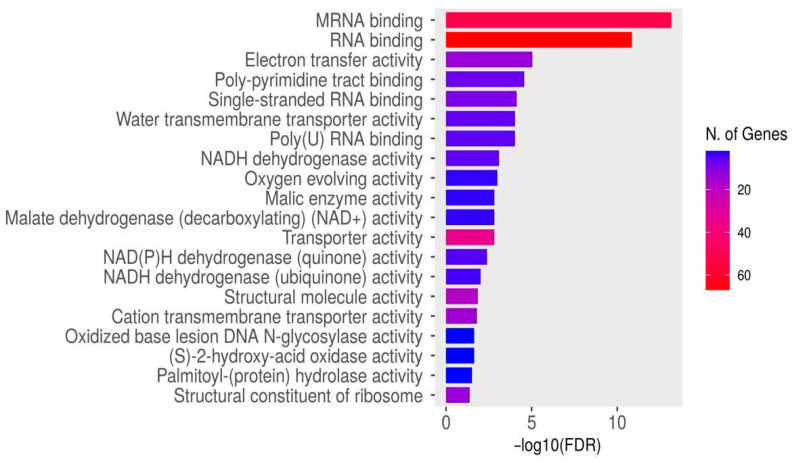
Functional enrichment analysis of the identified circRNAs in our data. The *x*-axis represents the log10 fold-change values, and the *y*-axis represents the gene functional categories. The diverse colors stand for distinct significance, and the various sizes indicate the diverse circRNAs numbers. Cluster colors are indicated for reference on the left side of the plot, along with the cluster gene numbers.

**Figure 17 life-15-01143-f017:**
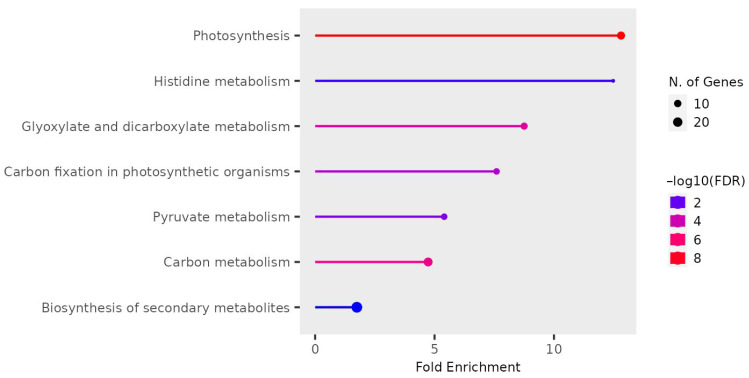
KEGG enrichment pathways of dysregulated circRNAs in our data. The *x*-axis represents the log10 fold-change values, and the *y*-axis represents the KEGG pathway categories. The diverse colors stand for distinct significance, and the various sizes indicate the diverse circRNA numbers. Cluster colors are shown for reference on the left side of the plot, along with the cluster gene numbers.

**Figure 18 life-15-01143-f018:**
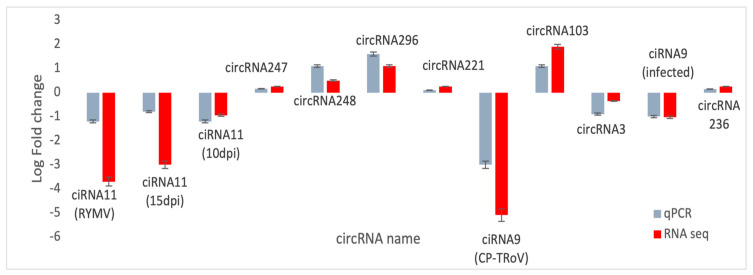
Validation of the selected dysregulated circRNAs from RNA-seq results was performed using qPCR. In the bar graphs, RNA-Seq data is represented in red, while qPCR data is indicated in blue. The error bars reflect the standard deviation (*n* = 2). The *y*-axis denotes the log2 fold-change for each gene in RT-qPCR and RNA-seq analysis. Each RNA sample originates from six different plants.

**Table 1 life-15-01143-t001:** Nucleotide sequences of forward (F) and reverse (R) primers used for confirmation of gene presence. F indicates forward primer; R indicates reverse primer.

Primer Name	Sequence (5′-3′)
TRoV CP-F	GAGGACCAATTCAGTGGTTACACC
TRoV CP-R	CTGCTGCCGTTGTTCCATCAGCGG
LTSV_sat_F	CCTACCATGGCCTCATCAGT
LTSV_sat_R	GCCGGTAGGATGATGGATTA
circ-LTSV-F	GGTCGACTCTAGAGGATCCCCCCCATGGCCTCATCAGT
circ-LTSV-R	GGCGCGGTCTAGATACGACTCACTATAGGGCGAATTCG
RYMV-F	ACAATTGAAG CTAGGAAAGG AGC
RYMV-R	CTCCCCCACC CATCCCGAGA ATT

**Table 2 life-15-01143-t002:** Bioinformatics software used.

Analysis Item	Software	Version
Quality control	FastQC	0.10.1
Adapter removal	Cutadapt	1.10
Genome mapping	Tophat	2.0.4
Back-splicing junction reads filter	Tophat-fusion	2.1
circRNA identification	CIRCexplore, CIRI	2.2.6, 2.0.2
Differential expression analysis	edgeR	NA
Interaction with miRNA	Targetmimics	NA

**Table 3 life-15-01143-t003:** Nucleotide sequences of forward (F) and reverse (R) primers used for confirmation of circRNA presence.

CircRNA ID	Reverse Primer	Forward Primer	Product Size
circRNA2	TGATTCCACTTCCTTCGATGC	ACTGTAATCGCGGCTTTGTC	142 bp
ciRNA33	TGAGCACATTGAACTATCCATGT	TACTCCTCCTGTTCGAACCG	115 bp
ciRNA65	CCTAATGTCAGGCTGTTGTTCT	TGATCTTTTCGTCCTATGAACCT	167 bp
ciRNA6	CCTAATGTCAGGCTGTTGTTCT	AGCAAACCTACGTCAAGAGAAC	135 bp
circRNA205	ACTCCCAAGCGCACAAATTC	TTTCTTCTTAGCGGCTTGGC	195 bp
ciRNA9	TGCTAATGTGCCTTGGATGA	TGCGTTCGGGAAGGATGAAT	125 bp
CircRNA105	GTAAAAGCAAGATGATACTTC	GGTGACACAAGGATTTTCAG	182 bp
ciRNA91	TCAGGCTGTTGTTCTCCTCTT	ACCAAAAGCAAACCTACGTCA	196 bp
ciRNA17	TGCTAATGTGCCTTGGATGA	TGCGTTCGGGAAGGATGAAT	188 bp
ciRNA3	ATTCGCGGGATTATGCACTC	GCATGAGAGCCAAATGAATCG	134 bp
circRNA296	TAGTACTTGCCTAGCGGACG	TGGTGGAGATTCTAGGCGAC	187 bp
circRNA248	TCCCTATCGAGTCAAAAGGAAGA	GTTCAGAACCGGATGATCAAGA	158 bp
circRNA61	TATGCCTGCTTTGACCGAGA	TGCAAGATCACGTCCCTCAT	157 bp
circRNA247	TGCACATAAGGCGTTTCATTT	GTGGCTATGGACGTTGCTAA	125 bp
CircRNA221	CCTTTCGGAGCTTCAACAGT	CTGGGATGATGAGGATGTGGA	165 bp
ciRNA8	ACTCCCTGCATTTAATTCCACT	GCGACTTGTCAGATATATCGGG	200 bp
Elongation factor alpha housekeeping gene (EF)			
Primer ID	Reverse primer	Forward primer	Product size
CC2012 (qRT-PCR)	GGTCTGCCTCATGTCCCTAA	TGGTGACGCTGGTATGGTTA	109 bp
EI382 (RT-PCR)	GCTCCTGGTCATCGTGATTT	GCTCCTGGTCATCGTGATTT	488 bp

## Data Availability

Data are contained within the article.

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
