# Peer review of "Profiling Plant circRNAs Provides Insights into the Expression of Plant Genes Involved in Viral Infection"

_life, 2025, doi:10.3390/life15071143_

Round 1
Reviewer 1 Report
Comments and Suggestions for Authors
The manuscript presents interesting findings on circRNA dynamics in A. thaliana under viral gene and non-host virus conditions. However, several issues require attention before the manuscript can be considered for publication.
It is recommended to provide more detailed information about the plasmids pBI121-CP-TRoV, pCambia1300-RYMV, and pCambia1300-scLTSV, including at least schematic diagrams of the sequences between T-DNA borders in the supplementary materials.
The supplementary material mentioned in line 244 could not be found and should be properly attached or referenced.
Specific Comments:
Line 9: Define "circRNAs" at first mention.
Line 10: Change “circular RNAs (circRNAs)” to “circRNAs”.
Line 17: Change “Turnip” to “turnip” and “Rice” to “rice”.
Line 18: Change “Lucerne Transient Streak Virus” to “Lucerne transient streak virus”.
Line 29: Modify “circular satellite RNA (scLTSV)” to “small circular satellite RNA of the Lucerne transient streak virus (scLTSV)”.
Line 51: Change “Turnip” to “turnip”.
Line 52: Change “Lucerne Transient Streak Virus” to “Lucerne transient streak virus” and “Rice” to “rice”.
Line 65: Change “the circular satellite RNA of the scLTSV virusoid” to “virusoid—scLTSV”.
Line 74: According to MDPI journal guidelines, subheadings should be numbered, and each major word should be capitalized. Please insert “2.1” at the beginning of this subheading and capitalize the first letter of each word.
Line 90: The reference to “Sambrook & William Russell, 2001” is missing in the References section; please add it.
Line 131: "Agrotransformation" can be interpreted as "Agrobacterium-mediated transformation," but it is not a standard or widely accepted English term. It is recommended to use the full term "Agrobacterium-mediated transformation" for clarity and correctness.
Line 132: Replace “Agrobacteria” with “Agrobacterium cells”.
Lines 145, 175, 199, 265, 266, 283, 305, 319, 330, 339, 341, 342, 344, 373: Replace “Arabidopsis” with "A. thaliana" (italicized).
Lines 149, 153: "PAGE" is a commonly used technique and does not require definition. However, if you prefer to define it, do so at its first occurrence in line 149, not line 153.
Line 154: The reference “Sambrook et al., 2001” is missing in the References section; please add it.
Line 163: Remove "viral/".
Lines 181, 392: Change “circular RNAs (circRNAs)” to “circRNAs”.
Line 185: E. coli should be italicized as it is a species name.
Lines 201, 220: De novo should be italicized, as it is a Latin term.
Tables: Add a table number, starting with “Table 1”, and adjust all table numbers accordingly. Ensure that the formatting is consistent with the table containing primer information.
Lines 229, 351: Define "capsid protein gene (CP)" at its first occurrence in line 137, not here.
Line 245: Remove the phrase “RT-PCR results”.
Line 255: The “800-base pair (bp) product” should be verified by Sanger sequencing and compared with the plasmid sequence used for transformation to confirm sequence integrity.
Line 280: What is the sequence identity between the 720 bp RT-PCR product from RYMV-infected rice plants and that from transgenic Arabidopsis plants? Please clarify.
Line 295: Merely observing the band size (522 bp and 844 bp) is insufficient for precise identification. Were these bands validated by Sanger sequencing? Please clarify. This concern applies to other gel images as well.
Line 370: Change “Rice yellow mottle virus (RYMV) capsid protein” to “RYMV CP”.
Line 404: Change “Arabidopsis thaliana” to "A. thaliana" and italicize.
Line 425: Figure 8 contains nine groups of data, but only five are labeled. Please annotate all groups clearly and completely.
Line 453: In Figure 9, annotate that "ChrC" refers to the chloroplast chromosome and "ChrM" refers to the mitochondrial chromosome.
Line 577: Provide detailed information about the primers used.
Line 592: Among the 760 differentially expressed circRNAs identified across the nine samples, are there any circRNAs induced by the inserted viral sequences? If so, can they still be classified as endogenous circRNAs?
Line 801: DOI information is missing from the references and should be provided where available.
Author Response
Dear Reviewer 1,
Thank you very much for taking the time and effort to review this manuscript. Please find the detailed responses in the attached file below and the corresponding revisions/corrections highlighted/in track changes in the re-submitted files. We hope that our manuscript will receive due consideration.

Reviewer 2 Report
Comments and Suggestions for Authors
In the submitted manuscript, Hashim et al.studied circRNA profiles in the Arabidopsis thaliana plants infected with host and non-host viruses. The presence of transferred genes in transgenic A. thaliana plants was confirmed by RT-PCR. The presence or absence of viral protein in transgenic A. thaliana was confirmed by Western analysis. Based on sequencing data from healthy, infected and transgenic Arabidopsis plants, 760 circRNAs were identified. Principal Component Analysis distinguished negative controls from the remaining samples. Most of the circRNAs were expressed in the chloroplast chromosome and only one in mitochondria. This association suggests a significant role for circRNAs in regulating critical photosynthetic processes. The results of the study suggest that transgenic expression of scLTSV in A. thaliana enhances resistance to TRoV and may be a new strategy for viral diseases. This study improves our understanding of the regulatory role of circRNAs in plant-virus interactions.
The Introduction section is currently missing.¾ of this section is a description of the study and only ¼ (8 lines) is a description of circRNAs without any references. This section should be completely rewritten.
The Bioinformatic analysis subsection is highlighted as a separate section, although it is part of the Materials and Methods section.
L.227. “Symptom development…”. The results of plant height measurements should be presented (L. 172).
L.241-243.“Detailed procedures for construct preparation, …, are provided in the Supplementary Section”. The Supplementary Section is missing in the manuscript.
Table 1 (sequences of primers) should be moved from Results to Materials and Methods.
There are 147 references in the References section, but they are not in the text .On the other hand, the Materials and Methods section contains references to Sambrook and Russell, 2001;Murashige and Skoog, but they are not in the References section.
Author Response
Dear Reviewer 2,
Thank you very much for taking the time and effort to review our manuscript. Please find the detailed responses below in the attached file and the corresponding revisions/corrections highlighted/in track changes in the re-submitted files. We hope that our manuscript will receive due consideration.

Reviewer 3 Report
Comments and Suggestions for Authors
The manuscript, titled "Profiling plant circRNAs provides insights into the expression of plant genes involved in viral infection," presents the results of a new line of research. The aim of the studies is to find out how the circular RNA profile of individual plants is affected by infection with various viruses. It is known that profiling plant circRNAs provides insights into the expression of plant genes involved in viral infection.
The introduction presents the significance of the topic and the research results to date in sufficient detail. The material and methods chapter is sufficiently detailed and well structured. It allows for the repetition of the studies and presents the applied molecular biological studies and bioinformatics procedures in sufficient detail.
The results are presented appropriately and well illustrated. The results highlight that in the test plant Arabidopsis thaliana the detected circRNA forms (288) are mostly associated with the chromosomes of chloroplasts. In this chapter, I propose to review the spelling of Arabidopsis thaliana. In some places, the scientific name is not in italics. It is also a formal error in the manuscript that in the case of Figure 8, the explanation of the figure slides to the next page.
The discussion chapter analyzes the achieved results in sufficient detail, but there is very little literature comparing similar results published in other research. This can also expand the currently modest number of cited references (19).
After making the minor corrections and additions suggested above, I recommend publishing the manuscript in the form of a scientific article.
Author Response
Dear Reviewer 3,
Thank you very much for taking the time and effort to review our manuscript. Please find the detailed responses below in the attached file and the corresponding revisions/corrections highlighted/in track changes in the re-submitted files. We hope that our manuscript will receive due consideration.

Reviewer 4 Report
Comments and Suggestions for Authors
The topic of plant circRNAs in viral infection is novel and the experimental design, including the use of both host (TRoV) and non-host (RYMV) viruses, and a circular satellite RNA (scLTSV), is comprehensive. The combination of deep sequencing, bioinformatic analyses, and biochemical assays (RT-PCR, Western blot) provides a multi-faceted approach. The finding that scLTSV can enhance resistance to TRoV is particularly noteworthy and suggests a promising avenue for plant viral resistance strategies.
Although, the topic is relevant, given the emerging role of circRNAs in plant stress responses, several concerns must be addressed before publication.
Even if the abstract and introduction apparently describe the objectives of the research, the “Results Obtained” segment seems like an extension of “Materials and Methods” rather than a demonstration of important results. The headings within the “Results” (e.g., “Constructs prepared in the current study”) are procedural and disturb the description flow.
Figures 1-5, although representing successful PCR and Western blot findings, are described rather disconnectedly. The “Confirmation of scLTSV (circ-LTSV) in transgenic A. thaliana infected with TRoV” and “Confirmation of scLTSV (circ-LTSV) in transgenic A. thaliana” appears inappropriate or should be united with clearer differences.
The preliminary “Results” part mainly approves the successful execution of molecular biology techniques (RT-PCR, Western blot, plant transformation). Though essential, these are mainly methodological confirmations. The fundamental innovative results associated with circRNA profiling and their functional implications (e.g., GO/KEGG analysis, ORFs in circRNAs, scLTSV-induced resistance) are only concisely stated in the abstract and then deferred to the “CircRNA sequencing” and succeeding segments.
The section on “Bioinformatic Analysis” efficiently describes the computational workflow, but it needs to benefit from a clearer link between the tools used and the specific biological questions they addressed.
The English language is normally clear and understandable, indicating a good understanding of scientific terminology. There are minor cases where phrasing should be enhanced for conciseness or natural flow, but these are not important enough to hinder understanding.
On the whole, the manuscript provides some important results and data and provides solution of an important research query. Addressing the above mentioned issues dealing with organization and flow of the results section, and minor linguistic improvements, it can be more suitable for publication. The new finding about scLTSVs part in improving resistance is mostly convincing and should be emphasized more obviously.
Comments on the Quality of English LanguageThe English language is normally clear and understandable, indicating a good understanding of scientific terminology. There are minor cases where phrasing should be enhanced for conciseness or natural flow, but these are not important enough to hinder understanding.
Author Response
Dear Reviewer 4,
Thank you very much for taking the time and effort to review our manuscript. Please find the detailed responses below in the attached file and the corresponding revisions/corrections highlighted/in track changes in the re-submitted files. We hope that our manuscript will receive due consideration.

Round 2
Reviewer 2 Report
Comments and Suggestions for Authors
The manuscript has been significantly improved.
L. 308-310.“Observations of plant morphology revealed notable differences between healthy and infected/transgenic plants” (Fig. 1).The presence/absence of significant differences between plant groups in Fig. 1 should be confirmed by statistical analysis (ANOVA).
L. 182, 183. Agrobacterium should be italicized.
Author Response
Reviewer #2
Reviewer comment:
L. 308-310. “Observations of plant morphology revealed notable differences between healthy and infected/transgenic plants” (Fig. 1).The presence/absence of significant differences between plant groups in Fig. 1 should be confirmed by statistical analysis (ANOVA).
Author rebuttal:
ANOVA and Tukey HSD statistical analysis have been duly performed and the significant differences between plant groups have been confirmed as included in the attached Excel file.
Reviewer comment:
L. 182, 183. Agrobacterium should be italicized.
Author rebuttal:
The term Agrobacterium has been italicized.
Reviewer 4 Report
Comments and Suggestions for Authors
The only persisting deficiencies are minor linguistic inaccuracies, which can be efficiently resolved during the final proofreading stage prior to publication, contingent upon acceptance.
Author Response
Reviewer #4
Reviewer comment:
The only persisting deficiencies are minor linguistic inaccuracies, which can be efficiently resolved during the final proofreading stage prior to publication, contingent upon acceptance.
Author rebuttal:
Linguistic inaccuracies have been corrected in the entire manuscript (as observed by “Track changes”).